# Production of Bio-Based Polyol from Coconut Fatty Acid Distillate (CFAD) and Crude Glycerol for Rigid Polyurethane Foam Applications

**DOI:** 10.3390/ma16155453

**Published:** 2023-08-03

**Authors:** Ma. Louella D. Salcedo, Christine Joy M. Omisol, Anthony O. Maputi, Dave Joseph E. Estrada, Blessy Joy M. Aguinid, Dan Michael A. Asequia, Daisy Jane D. Erjeno, Glenn Apostol, Henry Siy, Roberto M. Malaluan, Arnold C. Alguno, Gerard G. Dumancas, Arnold A. Lubguban

**Affiliations:** 1Center for Sustainable Polymers, MSU-Iligan Institute of Technology, Iligan City 9200, Philippines; malouella.salcedo@g.msuiit.edu.ph (M.L.D.S.); christinejoy.omisol@g.msuiit.edu.ph (C.J.M.O.); anthony.maputi@g.msuiit.edu.ph (A.O.M.); davejoseph.estrada@g.msuiit.edu.ph (D.J.E.E.); blessyjoy.aguinid@g.msuiit.edu.ph (B.J.M.A.); danmichael.asequia@g.msuiit.edu.ph (D.M.A.A.); daisyjane.erjeno@g.msuiit.edu.ph (D.J.D.E.); roberto.malaluan@g.msuiit.edu.ph (R.M.M.); arnold.alguno@g.msuiit.edu.ph (A.C.A.); 2Materials Science and Engineering Program, Graduate School of Engineering, MSU-Iligan Institute of Technology, Iligan City 9200, Philippines; 3Institute of Engineering and Computer Studies, Camiguin Polytechnic State College, Mambajao 9100, Philippines; 4Chemrez Technologies, Inc., Quezon City 1110, Philippines; gcapostol@chemrez.com (G.A.); henrysiy@chemrez.com (H.S.); 5Department of Chemical Engineering and Technology, MSU-Iligan Institute of Technology, Iligan City 9200, Philippines; 6Department of Physics, MSU-Iligan Institute of Technology, Iligan City 9200, Philippines; 7Department of Chemistry, The University of Scranton, Scranton, PA 18510, USA; gerard.dumancas@scranton.edu

**Keywords:** coconut fatty acid distillate, rigid polyurethane foam, bio-based polyols, high open-cell content foam, thermal insulating material

## Abstract

This study propounds a sustainable alternative to petroleum-based polyurethane (PU) foams, aiming to curtail this nonrenewable resource’s continued and uncontrolled use. Coconut fatty acid distillate (CFAD) and crude glycerol (CG), both wastes generated from vegetable oil processes, were utilized for bio-based polyol production for rigid PU foam application. The raw materials were subjected to catalyzed glycerolysis with alkaline-alcohol neutralization and bleaching. The resulting polyol possessed properties suitable for rigid foam application, with an average OH number of 215 mg KOH/g, an acid number of 7.2983 mg KOH/g, and a Gardner color value of 18. The polyol was used to prepare rigid PU foam, and its properties were determined using Fourier transform infrared spectroscopy (FTIR), thermogravimetric analysis/derivative thermogravimetric (TGA/DTA), and universal testing machine (UTM). Additionally, the cell foam morphology was investigated by scanning electron microscope (SEM), in which most of its structure revealed an open-celled network and quantified at 92.71% open-cell content using pycnometric testing. The PU foam thermal and mechanical analyses results showed an average compressive strength of 210.43 kPa, a thermal conductivity of 32.10 mW·m^−1^K^−1^, and a density of 44.65 kg·m^−3^. These properties showed its applicability as a type I structural sandwich panel core material, thus demonstrating the potential use of CFAD and CG in commercial polyol and PU foam production.

## 1. Introduction

The polyurethane (PU) industry is one of the chemical industries that remained heavily reliant on petroleum-derived products as raw materials [1]. A fundamental component in PU synthesis is a polyol, which is still widely produced from petroleum-based materials [2,3]. The global production of polyols alone reached approximately 22 million metric tons in 2019, amounting to USD26.2B, and placing it as the sixth most important polymer family [4]. Its global market is expected to grow up to USD34.4B by 2024. Being this contingent on non-renewable sources, the advancement of global industrialization of the PU industry is at risk [5]. The threat of depletion, economic volatility, and environmental impacts are the main factors that pushed the PU industry to search for new, renewable, and sustainable polyol raw materials [6].

One of the best routes of action is to utilize environmentally friendly products derived from renewable and natural molecules [7]. Vegetable oils (VOs) are among the most preferred substitute raw materials for polyols due to their abundant supply, low toxicity, biodegradability, inherent fluidity, and low cost [8]. This shift from using petroleum-based to bio-based polyols offers new options to valorize biomass into technical-grade polyols in the PU industry [4].

Most studies utilize VOs as the raw material for polyol production oils [9,10,11,12,13,14]. However, the direct utilization of these vegetable oils, particularly edible oils, in the PU industry is disadvantageous as it directly competes with the food and biofuel industries. In consequence, there is a need to develop more sustainable initiatives. One of the propitious options to address this concern is the direct utilization of vegetable oil byproducts from oil refineries. This action will lead to value-added products and advance the utilization of low-cost waste materials through a facile process.

One of the byproducts of the vegetable oil refining industry is fatty acid distillate (FAD). This material is a byproduct produced from the deodorization step, which is the final stage in the chemical refining of vegetable oils [15,16]. FAD is considered a low-value waste and is usually just disposed of, giving rise to environmental issues. This type of waste accounts for around 4% of the total composition of unrefined oil; thus, disposing of it is wastes potentially valuable material [16]. Although there are multiple uses for FAD, according to the literature, they are either expensive to execute or not well-established. Coconut oil is one such vegetable oil that can be a good source of coconut fatty acid distillate (CFAD). This material is not as widely used in the PU industry compared to other vegetable oils, such as palm, soybean, rapeseed, and castor oils [9,10,11,12,13,14]. This distillate contains a high amount of lauric acid along with a mixture of volatile organic compounds and impurities. This kind of fatty acid distillate has not been fully utilized yet in the PU industry as it has limited application due to the disadvantages of free fatty acids in polyols.

Another waste material from vegetable oils, particularly in the biodiesel industry, is crude glycerol (CG). CG is a major byproduct in biodiesel production teeming with impurities that hinders its direct industrial usage [17]. With this byproduct constituting 10% of the total biodiesel production, its direct utilization would be highly advantageous. Like FAD, CG’s existing applications mostly require high purity, thus necessitating a purification process. This limits the direct usage of CG in the PU industry since the presence of its impurities may compromise the properties of PU products.

The use of these two waste materials, CFAD and CG, diversifies the applications of vegetable oils without trading off their value in the food and biofuel industries. Thus, this study aims to devise a process centered around the glycerolysis reaction that will directly utilize CFAD and CG as starting materials for the synthesis of a CFAD-based polyol for PU foam applications. This study will also demonstrate the compatibility of the CFAD-based polyol as a precursor in producing a rigid thermal insulation PU foam.

Therefore, the current study directly utilizes CFAD and crude glycerol, both waste materials from agricultural products, in the production of polyol for rigid PU foam by integrating CFAD into crude glycerol, thereby yielding a fully bio-based polyol suitable for the targeted application. The effects of different process conditions during polyol synthesis on the acid and OH numbers of the CFAD-based polyol were studied and reported. Moreover, FTIR was used to investigate the structural configuration of the polyol. Further, the physical properties of the CFAD-based foam in terms of density, thermal conductivity, and open-cell content were determined. The compressive strength of the foam was also evaluated, and its molecular structure was inspected using FTIR. TGA and SEM were used to examine the foam’s thermal stability and morphology, respectively.

## 2. Materials and Methods

### 2.1. Materials

Coconut fatty acid distillate (CFAD) and crude glycerol (CG) were obtained from Chemrez Technologies, Inc. (Quezon City, Philippines). Their properties are shown in Table 1 and Table 2, respectively. VORANOL^®^ 490, DABCO^®^ 33-LV, POLYCAT^®^ 8, and DABCO^®^ DC 2585 were also obtained from the same facility. Potassium hydroxide, absolute ethanol, and hydrogen peroxide were purchased from Sigma-Aldrich (St. Louis, MO, USA).

### 2.2. Crude CFAD-Based Polyol Synthesis

The crude CFAD-based polyol was synthesized through glycerolysis with CFAD and CG. A 500 mL three-necked flask on a heating mantle equipped with a magnetic stirrer, thermometer, and condenser was used as the reacting vessel. The glycerolysis reaction was allowed to proceed at different CG loading ratios (0.5–3 molar ratio), reaction temperatures (170–190 °C), and reaction times (1–4 h). The materials with their designed ratio were loaded into the reaction vessel with 2 wt. % KOH catalyst and reacted with 600 rpm constant stirring.

### 2.3. CFAD-Based Polyol In Situ Treatment

The crude polyol produced from the glycerolysis of CFAD and CG was subjected to a two-step in situ treatment. The first step was neutralization, wherein a varying loading (8–14 wt. %) of an alkali-alcohol neutralizing agent was added to the reactor. The alkali-alcohol solution was produced by dissolving 8 g KOH in 100 g absolute ethanol. The neutralization reaction was conducted at 180 °C for 90 min, producing the neutralized polyol. The neutralized polyol was then subjected to the second part of the treatment process, i.e., bleaching, wherein hydrogen peroxide was loaded into the reactor vessel at 60 wt. % loading. The bleaching process was allowed to proceed at different bleaching temperatures (80–110 °C) and bleaching times (20–80 min). The final product, CFAD-based polyol, was then transferred to a separatory funnel to remove unreacted compounds.

### 2.4. PU Foam Preparation

A standard laboratory mixing and pouring procedure for making CFAD-based and petroleum-based (control) PU foams was used in this study. The rigid foam formulation used is summarized in Table 3. The B-side components were added into a 500 mL disposable plastic cup and mixed at 3450 rpm for 10–15 s. The mixture was then allowed to degas for 120 s.

The A-side component was added rapidly, stirring the mixture for 10–15 s at 3450 rpm. The mixture was poured immediately into a wooden mold (11.4 × 11.4 × 21.6 cm) with aluminum foil lining, and the foam was left to rise and cure at ambient conditions (23 °C) for 24 h.

### 2.5. Polyol and Foam Characterizations

The CFAD-based polyols were characterized for their acid number, OH number, and Gardner color index according to ASTM D1980 [18], ASTM D4274 Test D [19], and ASTM D1544 [20], respectively. A structural analysis was also performed using a Shimadzu IRTracer-100 FTIR spectrometer with the QATR-10 accessory (Shimadzu Corp., Kyoto, Japan) at a wavenumber range of 4000–500 cm^−1^ and 4 cm^−1^ resolution. The rigid PU foams were characterized by their mechanical and physical properties. The foam’s compressive strength was analyzed at 10% compression perpendicular to the direction of foam rise according to ASTM D1621 [21] using the AGS-X series universal testing machine (UTM) (Shimadzu Corp., Kyoto, Japan); in addition, the density was measured according to ASTM D1622 [22], and the thermal conductivity testing was performed according to ASTM C518 [23] using the FOX 200 heat flow meter (Laser-Comp, Wakefield, MA, USA) with a sample size of 150 × 150 × 20 mm. Lastly, the open/close cell content was measured according to ASTM D6226 [24] using the Quantachrome Ultrapyc 1200e automatic gas pycnometer (Germany). The CFAD-based foam was further characterized for its morphology using an analytical scanning electron microscope (SEM), namely JSM-6510LA from JEOL (Tokyo, Japan), and the molecular structure of the foam was also studied using FTIR. The foam’s thermal stability and degradation profile were studied using Shimadzu DTG-60H (Shimadzu Corp., Kyoto, Japan) at a temperature range of 50–750 °C, a heating rate of 10 °C/min, and a nitrogen flow rate of 40 mL/min.

## 3. Results and Discussion

This work reports on the conversion of coconut fatty acid distillate (CFAD) and crude glycerol (CG) to a polyol for rigid PU foam application through a three-step one-pot process: glycerolysis, neutralization, and bleaching. The glycerolysis of CFAD with crude glycerol (CG) converts the free fatty acids (FFA) in CFAD into mono-, di-, and triglycerides through an esterification reaction [25,26]. 

CG acts as the alcohol, providing the necessary hydroxyl groups for the reaction. Figure 1 shows the glycerolysis reaction. Only the monoglyceride and diglyceride have the hydroxyl functionalities needed during foam synthesis. Thus, it is essential to favor the OH-containing particles in the synthesis process.

### 3.1. Effect of Glycerolysis Conditions on Chemical Properties of Polyol

Figure 2 illustrates the effect of glycerolysis reaction time and temperature on the acid and OH numbers of the CFAD-based polyols. The results shown are for the polyols produced with a CFAD:CG ratio of 1:1. In addition, the data presented are with respect to the initial acid and OH number of CFAD. Figure 2a exhibits a downward trend in the acid number as the reaction proceeds, showing the consumption of CFAD. It can be deduced that a higher temperature favors the reaction, an observation that has similarities to the findings of Felizardo et al. (2011) [25]. A drastic decrease in the acid number of the polyols at temperatures 180 °C and 190 °C can also be noted until a reaction time of 3 h, after which the difference in acid number at the end of 4 h becomes negligible. In contrast, the reaction temperature of 170 °C shows a gradual decrease in the polyol’s acid number as the reaction proceeds. At the end of the reaction, the polyol still has an acid number of >100 mg KOH/g, which means that the reaction temperature is insufficient to promote glycerolysis. Consequently, this level has not been included in further sections of the study. These observations also agree with the literature stating that the reaction between carboxyl and hydroxyl groups occurs at elevated temperatures of 180–200 °C [27].

Additionally, the effects of glycerolysis time and temperature on polyol OH number were also investigated, as shown in Figure 2b. Both time and temperature influenced the polyol’s OH number. The initial OH number of CG at 497.50 mg KOH/g was lowered to approximately <200 mg KOH/g after the first hour of glycerolysis for both temperature levels. This may be attributed to the presence of primary hydroxyl groups in CG that are highly reactive. As the reaction proceeded, however, a slight increase in OH number was observed. After the 2-h mark, a sharp increase was recorded for 190 °C, while the 180 °C reaction maintained a steady incline upwards. This suggests the start of fatty acid cleavage from the polyol [28]. As such, this phenomenon is more drastic at higher temperatures. Thus, the condition determined to be the most appropriate for this application is glycerolysis at 180 °C for 2 h.

### 3.2. Effect of Crude Glycerol Loading

The effect of crude glycerol (CG) loading was also investigated to determine the most suitable polyol formulation. Figure 3 shows the acid and OH numbers behavior of the polyol synthesized at 180 °C for 2 h at different CG loadings. The results show that increasing the CG loading tends to decrease the product’s acid number due to increased available OH that can react with CFAD. In addition, increasing the CG loading increases the OH number of the polyol due to an increase in excess OH present in the mixture. These trends imply a successful esterification between CFAD and CG, forming a combination of the three products listed in Figure 1. However, a slight increase in acid number was recorded at a CG loading of 3 molars. A slight decrease in OH number was also observed at this point. This phenomenon may be due to an oxidation reaction of the excess OH groups in CG, wherein OH groups are oxidized to form carboxyl groups, thus leading to an increase in acid and a decrease in OH numbers [29,30,31].

### 3.3. Effect of Alkali-Alcohol Loading

The primary goal of neutralization, the first step of the in situ treatment, is the removal of excess free fatty acids (FFA) after the glycerolysis reaction. Excess FFA contributes to the residual acidity of the polyol, which is detrimental to the polyol’s reactivity during the PU foaming process [32]. In addition, the presence of these FFA in the polyol poses negative effects on its storage stability, resulting in poor color and odor. To compensate, a neutralization reaction is essential to improve the polyol’s quality and application. The neutralization step uses an alkali-alcohol solution composed of KOH and ethanol, wherein FFAs react with the base and form the soap stock, as shown in Equation (1) [15]. This reaction lowers the acid number of the polyol through the consumption of the FFAs. The produced soap is generally insoluble in the oil-based polyol and can be easily removed via dissolution in a solvent and subsequent mechanical separation based on the difference in specific gravity [33]. The alcohol component in the alkali-alcohol solution serves as the solvent that dissolves the soap stock and facilitates the separation of the polyol. Figure 4 illustrates the effect of alkali-alcohol loading on the acid number of the CFAD-based polyol. It can be observed that there is a significant reduction in the polyol’s acid number, as predicted, due to the consumption of excess FFA. The highest acid reduction was recorded with an alkali-alcohol loading of 10%. Beyond this point, the acid number showed a significant increase. This may be due to the production of emulsified polyol or salt formation induced by the emergence of potassium carboxylate salts destabilized by ethanol, as previously determined by a related study by Xu et al. (2001) [34]. Thus, 10% alkali-alcohol loading is employed in the subsequent areas investigated in this study.
R–COOH (acid) + KOH (base) → R−COOK (soap) + H_2_O,(1)

### 3.4. Effect of Bleaching Conditions

The bleaching process is employed mainly to improve the color of the CFAD-based polyol. Hydrogen peroxide (6%) was used as the bleaching agent in accordance with the process performed by Frey et al. (2012) [35]. The color change was characterized by the Gardner color index of the samples. Table 4 lists the color indices of the products and the reactants. Evidently, the bleaching process improved the CFAD-based polyol’s appearance compared to the crude polyol’s color index, with the latter having an index of 35 and the former having an index of 18. With respect to its raw materials, specifically CFAD, there is also a significant improvement in its color.

Aside from color improvement, bleaching can also remove other minor components in the polyol, such as FFA and non-fatty materials as well as the components that give an odor to the polyol [36]. In consequence, the effects of bleaching temperature and time on the properties of the CFAD-based polyols were investigated using crude polyol. This is performed consecutively after the neutralization step. There is a need to study these factors as bleaching using peroxide is highly subjective and requires customization for different polyols to achieve optimal decolorization and minimal degradation effects [35]. Figure 5 shows the impact of bleaching temperature on the acid and OH numbers of the polyols, while Figure 6 depicts the influence of bleaching time on the polyol properties. Both the acid and OH values of the polyols were affected by the two factors in varying degrees. The acid value is significantly lowered with increasing bleaching temperature and time, as shown in Figure 5a and Figure 6a, respectively. In both setups, there is a decreasing trend in the acid number of the polyol as hydrogen peroxide is consumed in the bleaching reaction. However, an increase in acid number is observed at temperatures above 100 °C and reaction times beyond 1 h.

On the other hand, the OH number drastically decreased after the temperature of 100 °C and time of 1 h, as shown in Figure 5b and Figure 6b. These findings are similar to the findings of Kamairudin et al. (2021) [37]. It can be deduced from the behavior of the polyol’s acid and OH numbers that oxidation occurred in the reaction due to the addition of peroxide as the bleaching agent.

### 3.5. Mechanical and Physical Analyses of Rigid PU Foams

Being a cellular material, the most common application of rigid PU foams is against compressive loadings [38,39]. When coupled with low thermal conductivity for insulation functionality, the use and potential of PU foams in construction become interminable [40]. Thus, the two main areas of investigation focused on here are the mechanical property of the foam, specifically, compressive strength, and the physical properties, which include density, thermal conductivity, and cell type.

The mechanical and physical properties of the CFAD-based foam are listed in Table 5, along with the properties of petroleum-based control foam and standard values of rigid, cellular materials for type I structural sandwich panel core applications according to ASTM E-1730 [41]. The compressive strength of the CFAD-based foam is recorded to be 210.43 kPa, which is well within the range of type I sandwich panel core insulators that require only a minimum compressive strength of 137.9 kPa [41]. An important thing to note is that this result is competitive for a material with bio-replacement. This finding can be compared to a similar study with the same polyol replacement, which only yielded rigid foams with compressive strength of 17–32 kPa [42]. Another crucial property of rigid foams is their density. The CFAD-based foam recorded a density of 44.65 kg/m^3^. Compared with the standards, the obtained result is slightly greater than the required foam density of ≤38.4 kg/m^3^ for the type I structural sandwich panel core [41]. Despite this, commercially available products of similar applications have densities between 30 and 45 kg/m^3^ [43], showing that the CFAD-based PU foam satisfies the requirements of materials for this application.

It is important to note that the compressive property of the CFAD-based foam at 210.43 kPa decreased significantly compared with the control foam at 768.09 kPa. However, this decrease was also accompanied by a considerable difference in the densities of the foams (77.96 kg/m^3^ for the control foam). This difference in behavior between the mechanical and physical properties of the foams can be explained by the direct correlation between foam density and compressive strength [42,44,45,46]. This correlation states that the compressive strength of the foam tends to increase with increasing density. The addition of the CFAD-based polyol resulted in a foam with low density. Thus, its compressive strength also decreased. These results may be due to several factors commonly affecting bio-based PU foams. A primary factor is the lower functionality of the CFAD-based polyol with only an OH number of 215 mg KOH/g compared with 490 mg KOH/g of the petroleum-based polyol (VORANOL^®^ 490). A lower OH number leads to lower crosslinking densities between the polyol and isocyanate that constitutes the hard segments of the foam matrix [47]. Low crosslinking leads to low apparent densities of the resulting foam and, thus, will have low compressive strength. Moreover, the addition of CFAD-based polyol led to a drastic increase in the open cell content of the foam from 11.50% of the control foam to 92.71% of the CFAD-based foam. This also contributes to the lower physical and mechanical properties of the CFAD-based foam. High foam open-cell content was also observed in a similar study, wherein the addition of vegetable oil-based polyols increased the open-cell content of the resulting foams [42]. A possible explanation of this phenomenon is the incompatibility of the CFAD-based and petroleum-based polyol blend that resulted in weak polymer membranes between pores, causing the cells to open.

As for the main evaluating criteria for a material’s capability to be a thermal insulator, the thermal conductivity of the foam is the most straightforward indicator. Although indicated standard values in Table 5 show thermal conductivity values ≤ 36 mW·m^−1^K^−1^ [41], typical thermal conductivity values of thermal insulators at ambient conditions are between 20 and 60 mW·m^−1^K^−1^ [48]. The thermal conductivity recorded for the rigid foam block in this study was 32.10 mW·m^−1^K^−1^, while that of the control foam was 33.01 mW·m^−1^K^−1^. The insulative property of the CFAD-based foam slightly improved compared with the control foam, and it satisfied the standard value for type I sandwich panel insulation foam. Moreover, it also satisfies the industrially acceptable thermal conductivities of insulating materials, and can be considered superior to other studies that yielded vegetable oil-based foams with thermal conductivities of 39–52 mW·m^−1^K^−1^ [42,49].

### 3.6. FTIR Analysis of CFAD-Based Polyol and Rigid PU Foams

Figure 7 exhibits the FTIR spectra of the raw materials, CFAD and crude glycerol, and the untreated and treated polyol products. One of the main features of the raw materials is the absence of an O-H peak of CFAD between 3700 cm^−1^ and 3200 cm^−1^ in contrast to the broad peak observed in crude glycerol. In the polyol products, the O-H peak decreased. This suggests the occurrence of an esterification reaction between the OH groups of crude glycerol and the carboxylic acid groups in CFAD. Moreover, between the untreated and treated CFAD-based polyols, the O-H peak is significantly pronounced in the latter. These peaks are much more discernible on the polyol products than on the reactants. This suggests the occurrence of the esterification reaction of crude glycerol with CFAD. In addition to this, the sharp peak at 1750 cm^−1^ is an indication of C=O ester stretching, which further supports the proposed esterification reaction between CFAD and crude glycerol.

The IR spectrum of the foam was also investigated. Figure 8 depicts the key transmission bands on the spectrum of the rigid PU foam synthesized using the CFAD-based polyol. The detected sharp peak at 3312 cm^−1^ corresponds to the N-H stretching of the urethane bond. The low intensity of this band is characteristic of rigid foams as it signifies little to no presence of excess OH groups [50]. This suggests a high degree of crosslinking through the reaction of OH groups, thus increasing the foam’s rigidity. Correspondingly, the weak bands observed at 2925 cm^−1^ and 2860 cm^−1^ represent the C-H stretching of groups. Specifically, these groups represent the incorporated fatty acid chains in the glycerol backbone of the polyol. The peak at 1705 cm^−1^ signals the presence of C=O from the free urethane groups and the C=O from the fatty acid side chains of the CFAD-based polyol. Lastly, the conspicuous peak at 1514 cm^−1^ shows the C-N stretching band of the urethane linkage.

### 3.7. Foam Morphology and Cell Type

The morphology of the CFAD-based foam is shown in Figure 9. An interesting observation of the scanned image is the type of cell dominating the CFAD-based foam structure, as examined using the SEM. Evaluation of the foam morphology reveals the PU foam matrix’s open- and closed-cell structures, with most of the structure revealing an open-celled network. This observation was confirmed by the 92.71% open-cell content of the foam tested using a gas pycnometer, as listed in Table 5. Cell regularity is moderately observable from the morphology presented, with seemingly hexagonal cellular formation accompanied by occasional cellular breaks along the cell boundaries. Microscale level evaluation of the matrix revealed a consistent thin-film-like appearance, affirming not only crosslinking of the −OH moieties with the −NCO of the isocyanate but also with the dangling chains caused by the fatty-acid tails of CFAD-based polyol. This combined cross-linking may result in increased compressive strength of the resulting foam material, although with a lower density and high open-cell content. Given this level of open-cell content, the foam’s compressive strength at 210.43 kPa can be considered impressively high since the higher the open-cell content, the lower the material’s compressive strength [42]. Generally, open-celled foams have a compressive strength of approximately 10 kPa. Moreover, this feature constitutes the most insulating property of the foam as it has fewer PU linkages that decrease thermal conductivity.

### 3.8. Thermal Analysis of Rigid PU Foams

TGA/DTA was used to determine the thermal stability of the synthesized PU foam made from CFAD. The thermal decomposition of PU foam samples was conducted in an N2 environment at a heating rate of 10 °C/min. The TGA/DTA curve of the synthesized PU foam can be seen in Figure 10. The first stage of degradation (T_m1_) is related to the degradation of PU soft segments in the case of PU foam and the decoupling and pyrolysis of fatty acid dangling chains [40]. The second stage of decomposition (T_m2_) is attributed to the degradation of the urea and carbodiimide group and the isocyanurate rings [51]. Lastly, the third decomposition stage (T_m3_) is associated with the fracture of the polyester and the crosslinked [52]. The corresponding weight loss in % per degradation is shown in Table 6.

## 4. Conclusions

This study directly used coconut fatty acid distillate (CFAD) and crude glycerol (CG) to synthesize polyol via glycerolysis for rigid PU foam. The appropriate glycerolysis conditions were determined to be a CFAD:CG ratio of 1:2, a reaction temperature of 180 °C, and a reaction time of 2 h. Additional steps were also employed to improve the properties of the CFAD-based polyols: alkali-alcohol neutralization and hydrogen peroxide bleaching. Investigation of these processes showed that an alkali-alcohol loading of 10% and bleaching conditions of 100 °C and 1h yielded a suitable CFAD-based polyol for rigid foam application. Consequently, the CFAD-based polyol was used as a partial substitute for petroleum-based polyol. A 20% CFAD-based polyol loading in the foam formulation produced a less dense CFAD-based rigid foam than petroleum-based foam. The former, with a density of 44.65 kg·m^−3^, has a compressive strength of 210.43 kPa. Similar to the petroleum-based foam, the properties of the CFAD-based foam conformed with the direct correlation between density and compressive strength in PU foams. The expectedly lower density and compressive strength of the CFAD-based foam are due to the lower OH number of the CFAD-based polyol that led to lower crosslinking density in the foam matrix. Moreover, the CFAD-based foam has an excellent thermal conductivity of 32.10 mW·m^−1^K^−1^ and a high open-cell content of 92.71%, supported by its morphology. These mechanical and physical properties showed that the CFAD-based rigid PU foam could be used as a type I structural sandwich panel core insulating material in wall and roof panels in industrial, commercial, and residential settings. Thus, this novel study has demonstrated the potential of directly utilizing CFAD and CG as polyol raw materials and serves as starting ground for future work geared towards improving the applicability of these materials in the PU industry.

## 5. Patents

A patent entitled “Coconut fatty acid distillate-based polyols,” resulting from this work, with application number 1/2022/050672, has been filed and submitted to the Intellectual Property Office of the Philippines (IPOPHL).

## Figures and Tables

**Figure 1 materials-16-05453-f001:**
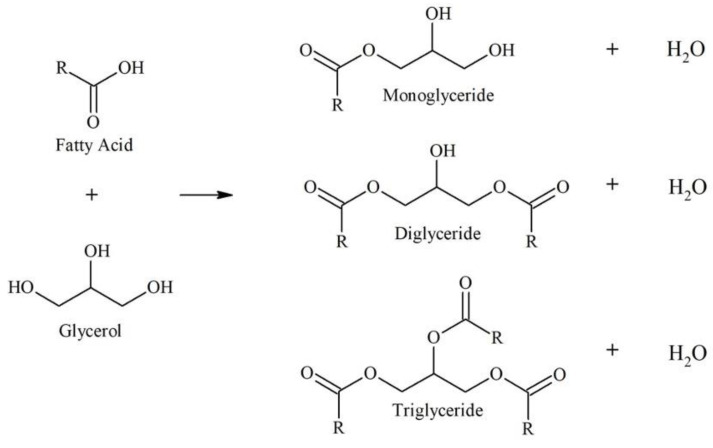
Glycerolysis of free fatty acid in coconut fatty acid distillate with glycerol producing mono-, di-, and triglycerides. Adapted from Felizardo et al. (2011) [25] and Mamtani et al. (2021) [26].

**Figure 2 materials-16-05453-f002:**
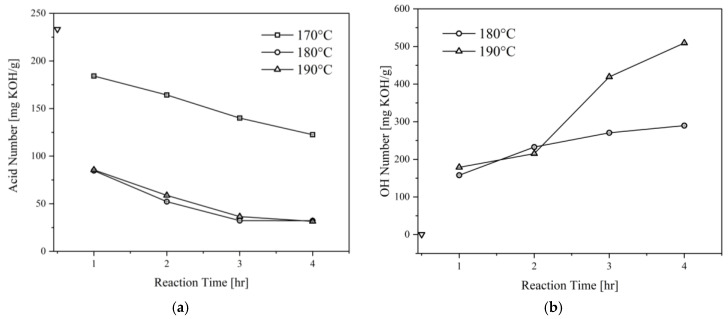
Effect of reaction time and temperature on the (**a**) acid number and (**b**) OH number of polyols produced from coconut fatty acid distillate (CFAD) and crude glycerol (CG) at CFAD:CG ratio of 1:1. The symbol ∇ refers to the initial properties of CFAD.

**Figure 3 materials-16-05453-f003:**
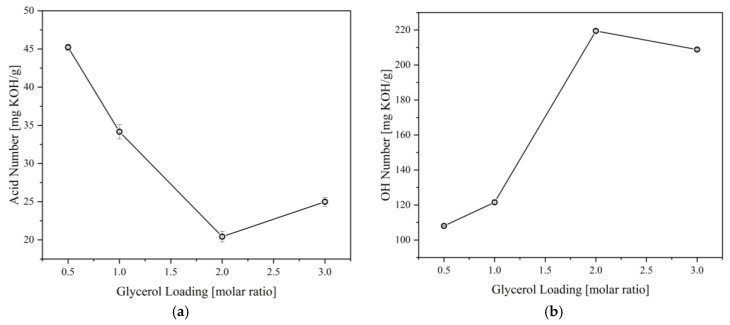
Effect of glycerol loading on the (**a**) acid number and (**b**) OH number of polyols produced from coconut fatty acid distillate (CFAD) and crude glycerol (CG) at 180 °C for 2 h.

**Figure 4 materials-16-05453-f004:**
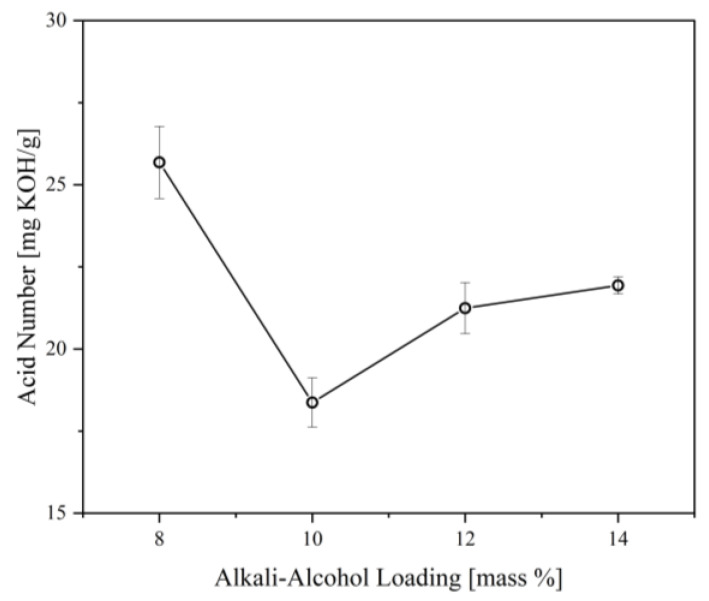
Effect of alkali-alcohol loading on the acid number of polyol synthesized from coconut fatty acid distillate (CFAD) and crude glycerol (CG) at CFAD:CG ratio of 1:2 at 180 °C for 2 h.

**Figure 5 materials-16-05453-f005:**
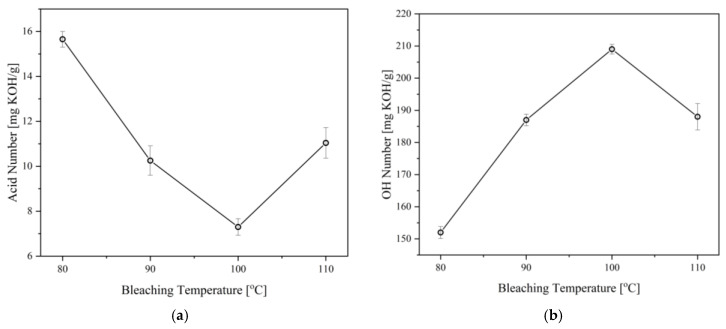
Effect of bleaching temperature on the (**a**) acid number and (**b**) OH number of polyols produced from coconut fatty acid distillate (CFAD) and crude glycerol (CG) at CFAD:CG ratio of 1:2, 180 °C for 2 h, neutralized with 10% alkali-alcohol loading, and bleached for 1 h.

**Figure 6 materials-16-05453-f006:**
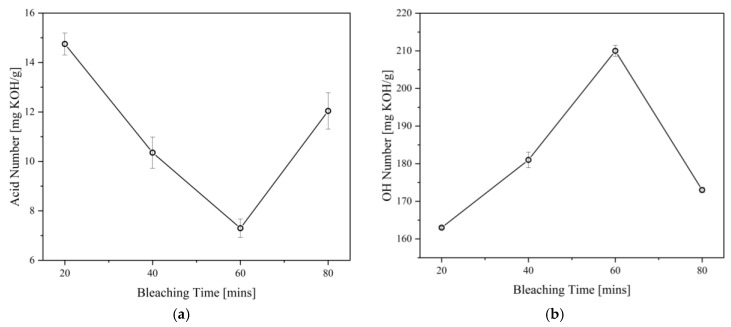
Effect of bleaching time on the (**a**) acid number and (**b**) OH number of polyols produced from coconut fatty acid distillate (CFAD) and crude glycerol (CG) at CFAD:CG ratio of 1:2, 180 °C for 2 h, neutralized with 10% alkali-alcohol loading, and bleached at 100 °C.

**Figure 7 materials-16-05453-f007:**
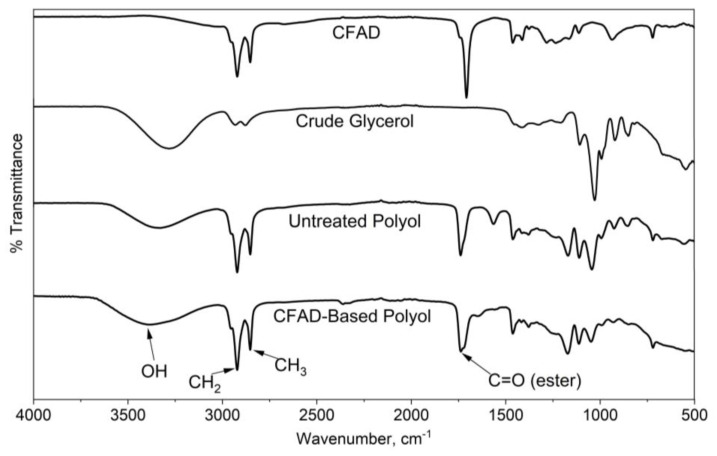
Fourier transform infrared (FTIR) spectra of coconut fatty acid distillate (CFAD), crude glycerol (CG), crude polyol, and CFAD-based polyol.

**Figure 8 materials-16-05453-f008:**
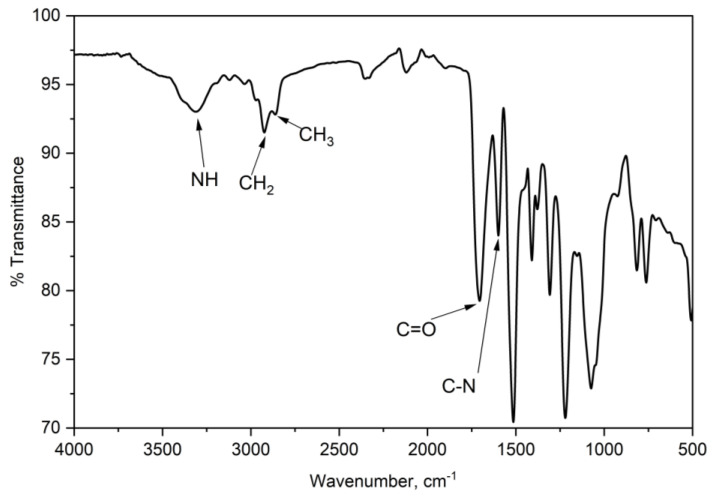
Fourier-transform infrared (FTIR) spectra of rigid polyurethane foam synthesized using coconut fatty acid distillate (CFAD)-based polyol.

**Figure 9 materials-16-05453-f009:**
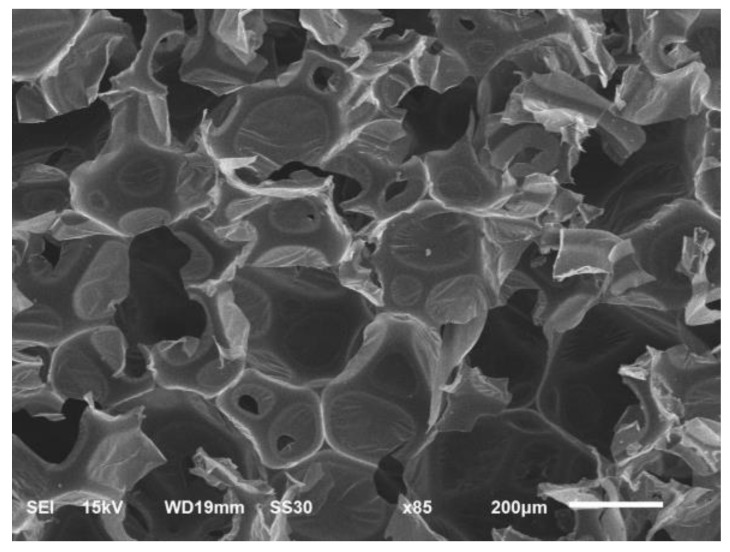
Morphology of the rigid polyurethane foam from coconut fatty acid-based polyol.

**Figure 10 materials-16-05453-f010:**
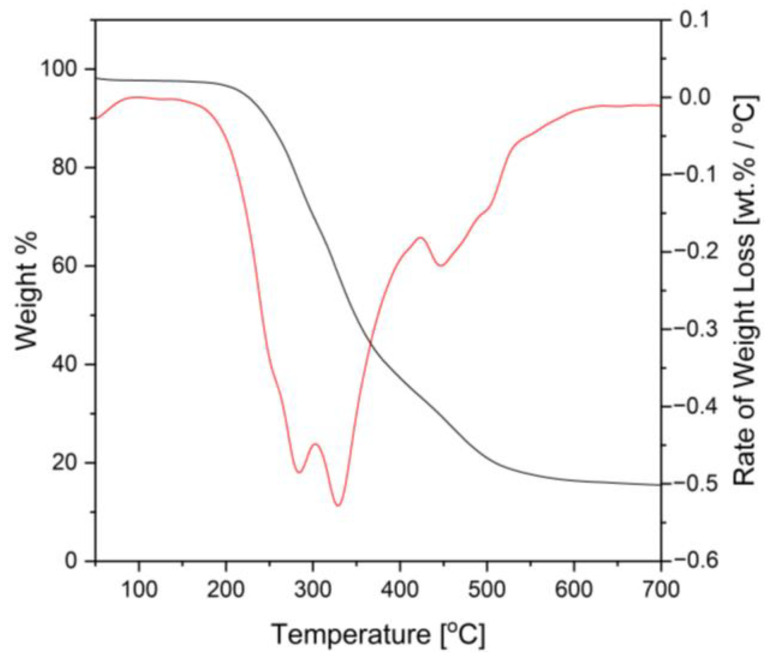
Thermogravimetric analysis (TGA) and differential thermal analysis (DTA) graphs of rigid polyurethane foam from coconut fatty acid distillate (CFAD)-based polyol.

**Table 1 materials-16-05453-t001:** Coconut fatty acid distillate (CFAD) properties.

Parameters	Test Method	CFAD Properties
Acid Number, mg KOH/g	ASTM D1980 [18]	233.33
OH Number, mg KOH/g	ASTM D4274 [19]	nil
Iodine Value, g I_2_/100 g	AOCS Cd 1-5	14.6
Saponification Value, mg KOH/g	AOCS Cd 3-25	257.56
Appearance	Ocular Inspection	Clear, dark yellow to brown

**Table 2 materials-16-05453-t002:** Crude glycerol (CG) properties.

Components	CAS Number	CG Properties
OH Number, mg KOH/g	56-81-5	497.50
Water	7732-18-5	12% max
Methanol	67-56-1	0.50% max
Fatty Acid and Ester	67762-38-3	0.10% max
Ash	-	8% max

**Table 3 materials-16-05453-t003:** CFAD-based and petroleum-based (control) rigid polyurethane foam formulation.

Foam Components	Parts by Weight	
	Control Foam	CFAD-Based Foam
B-side Materials		
VORANOL^®^ 490	100	80
CFAD-based Polyol	0	20
DABCO^®^ 33-LV	0.25–0.75	
POLYCAT^®^ 8	1.0–1.5	
DABCO^®^ DC 2585	0.5–1.0	
Distilled Water	0	
A-side Materials		
Polymeric MDI	Index 110	

**Table 4 materials-16-05453-t004:** Gardner color index of coconut fatty acid distillate (CFAD)-based polyols in comparison with raw materials and untreated crude polyol.

Sample	Gardner Color Index
CFAD-based polyol	18
Crude polyol	35
CFAD	27
Crude glycerol	6

**Table 5 materials-16-05453-t005:** Mechanical and physical properties of rigid polyurethane foam from coconut fatty acid distillate (CFAD)-based polyols compared with a petroleum-based control foam and standard values for type I rigid foam for structural sandwich panel core application.

Properties		Control Foam	CFAD-Based PU Foam	Standard Values for Type I Sandwich Panel Core [41]
Mechanical	Compressive Strength, kPa	768.09 (±7.15)	210.43 (±4.67)	≥137.9
Physical	Density, kg·m^−3^	77.96 (±0.53)	44.65 (±3.74)	≤38.4
	Thermal Conductivity, mW·m^−1^ K^−1^	33.01 (±1.37)	32.10 (±0.26)	≤36
	Open Cell Content, %	11.50 (±0.91)	92.71 (±0.39)	-

**Table 6 materials-16-05453-t006:** Thermal decomposition temperatures and percent weight loss of coconut fatty acid distillate (CFAD)-based polyurethane foam.

**Thermal Decomposition**	**T_m1_ (°C)**	**T_m2_ (°C)**	**T_m3_ (°C)**
	282.32	327.25	459.70
**% Weight Loss**	9.72	51.97	22.08

## Data Availability

The data presented in this study are available on request.

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
