# Peer review of "Production of Bio-Based Polyol from Coconut Fatty Acid Distillate (CFAD) and Crude Glycerol for Rigid Polyurethane Foam Applications"

_materials, 2023, doi:10.3390/ma16155453_

Round 1

Reviewer 1 Report

The manuscript with title Production of bio-based polyol from coconut fatty acid 2 distillate (CFAD) and crude glycerol for rigid polyurethane 3 foam applications. The idea is good and results are very impressive but i have only few question.

1. Where applicability of this matterials?

2. Is it economical? ecofriendly?

3.  Is it commercially feasible at large scale production and uses?

Its good 

Reviewer 2 Report

Salcedo and co-authors reported the synthesis of bio-based polyols from coconut fatty acid distillate (CFAD) and crude glycerol, mainly discussed the effect of process parameters on the acid number and OH number of polyols. Also, the CFAD-based polyol was used to prepare rigid polyurethane foam. Overall, some valuable results have achieved, however, there are some problems that should be clarified before further consideration. For improvement, as a reference, reviewer raised the following questions:

1. The author stated that “This study propounds a sustainable alternative to petroleum-based polyurethane (PU) foams aiming to curtail this nonrenewable resource's continued and uncontrolled use.” However, one can see from Table 3 that the polyols used for PU foams are mainly petroleum-based (VORANOL® 490, 80 wt%) rather than the synthesized CFAD-based (only 20 wt%). From this point, the importance and significance of this work is not clear. What if increasing the loading of CFAD-based polyol?

2. How about the effect of partial substitution of VORANOL® 490 with CFAD-based polyol on the foam-forming, mechanical and physical related properties? The author should compare it with the petroleum-based (VORANOL® 490 as polyol) PU foam, or even those with different loading of CFAD-based polyol.

3. In Table 5, the statement of “Standard Values” is not accurate. Those are reported values from previous work. Also, error estimates should be included in Table 5.

4. Section 3.7 “FTIR Analysis of CFAD-based Polyol and Rigid PU Foams” can be discussed before section 3.6 “Foam Morphology and Cell Type”.

5. When discussing the interaction of CFAD-based Polyol in PU foam by FTIR, the author need to consider both N-H stretching and C=O region. Details may refer the reported literature, e.g. Journal of Colloid and Interface Science, 2022, 621, 385-397.

6. The ordinate values in Figure 10 should be corrected.

7. In ABSTRACT, an abbreviation of FTIR, TGA/DTA, and UTM are firstly used, please give its full name.

Reviewer 3 Report

Please provide explanations to the following questions and ADD your responses to the manuscript in their proper parts.   

1.       How did the authors address CFAD and crude glycerol impurities with varying compositions and achieve a standardized composition for reliable production? 

2.       How did the authors optimize the reaction kinetics, such as the reaction rate and conversion efficiency to ensure a high yield of polyol while minimizing unwanted byproducts?

3.       After the reaction, the resulting mixture needs to be purified and separated to isolate the desired polyol. How did the authors employ efficient separation techniques to achieve a high-purity polyol in the presence of other compounds and impurities?

4.       How did the authors ensure that the polyol maintains its desired properties over time and does not degrade or undergo unwanted reactions?

5.       Authors claim that their proposed production process can be scaled up for industrial for rigid polyurethane foam applications but did not provide any raw material availability, process optimization, energy consumption, and overall economic viability. Please elaborate more on these terms.

Mentioned above.

Round 2

Reviewer 2 Report

The revised manuscript has addressed some of my concerns. However, the foam-forming, mechanical and physical related properties of CFAD-based polyol were not compared with the petroleum-based (VORANOL® 490 as polyol) PU foam. Although the authors stated that "The main significance and novelty of the study in is the direct utilization of CFAD and CG in producing a polyol that conforms to the requirements of rigid PU foam application in terms of acid and OH numbers". Considering the low loading of CFAD-based polyol (only 20 wt%), it is hard to reach that conclusion. If the mechanical and physical related properties of PU foam with 20 wt% CFAD-based polyol decrease, what’s the meaning of this work?

Reviewer 3 Report

N/A

As mentioned above.

Author Response

Point 1: Does the introduction provide sufficient background and include all relevant references? (Can be improved)

Response 1: The introduction sections were edited to reflect some information that highlights the content of the study and the application to which the product of the study may be used.

Point 2: Are all the cited references relevant to the research? (Can be improved)

Response 2: The authors added new references, which we believe further propels the present work's significance. Some references were also removed, which did not fit the information of the revised discussions. All the remaining cited references were deemed relevant by the authors as they support the discussions presented in this work. 

Point 3: Is the research design appropriate? (Must be improved)

Response 3: The research design was modified to include the synthesis and analysis of petroleum-based PU foam as a control material with which the CFAD-based foam properties were compared. The authors believe that this comparison, as highly suggested by the reviewers, painted a better picture of the effect of the novel CFAD-based polyol on the mechanical and physical properties of the resulting foam.

Point 4: Are the methods adequately described? (Must be improved)

Response 4: The Materials and Methods section was revised to give a more precise description of the methods employed in the study in order to be replicable for other researchers and authors that would want to investigate the same area of concern as the present work.

Point 5: Are the results clearly presented?

Response 5: Minor editing was done in the entirety of the paper to improve further the presentation of results and ideas and to correct typographical errors.

Point 6: Are the conclusions supported by the results? (Can be improved)

Response 6: The Conclusions section of the paper was edited to cater to the changes in the results discussed in the Results and Discussions section. The authors highlighted the most significant results in the present work and its potential impact on the PU industry.